# Dietary Docosahexaenoic Acid (DHA) and Eicosapentaenoic Acid (EPA) Operate by Different Mechanisms to Modulate Hepatic Steatosis and Hyperinsulemia in *fa/fa* Zucker Rats

**DOI:** 10.3390/nu11040917

**Published:** 2019-04-24

**Authors:** Lena Hong, Peter Zahradka, Luis Cordero-Monroy, Brenda Wright, Carla G. Taylor

**Affiliations:** 1Department of Food and Human Nutritional Sciences, University of Manitoba, Winnipeg, MB R3T 2N2, Canada; lenahong_@hotmail.com (L.H.); pzahradka@sbrc.ca (P.Z.); lcordero-monroy@sbrc.ca (L.C.-M.); 2Canadian Centre for Agri-Food Research in Health and Medicine, St. Boniface Hospital Research Centre, Winnipeg, MB R2H 2A6, Canada; bwright@sbrc.ca; 3Department of Physiology and Pathophysiology, University of Manitoba, Winnipeg, MB R3E 0J9, Canada

**Keywords:** n3-fatty acids, eicosapentaenoic acid, docosahexaenoic acid, α-linoleic acid, hepatic steatosis, inflammation, *fa/fa* Zucker rats

## Abstract

Hepatic steatosis, an early stage of non-alcoholic fatty liver disease, is commonly present in obesity and type 2 diabetes, and is associated with reduced hepatic omega-3 polyunsaturated fatty acid (n3-PUFA) status that impacts on the anti-inflammatory and insulin sensitizing functions of n3-PUFA. Our objective was to directly compare plant- and marine-based n3-PUFA (α-linoleic acid (ALA)), eicosapentaenoic acid (EPA), and docosahexaenoic acid (DHA)) for their effects on hepatic steatosis, markers of hepatic inflammation and fibrosis, and insulinemia in obese rats. *Fa/fa* Zucker rats were provided diets containing ALA, EPA, DHA, or linoleic acid (LA, n6-PUFA) for eight weeks and compared to baseline *fa/fa* rats and lean Zucker rats fed LA-rich diet for eight weeks. Both DHA and EPA groups had liver lipid similar to baseline, however, DHA was more effective than EPA for reducing hepatic fatty acid synthase (FAS), increasing the proportion of smaller lipid droplets, reversing early fibrotic damage, and reducing fasting hyperinsulinemia. EPA was more effective for reducing FoxO1. Dietary ALA did not attenuate hepatic steatosis, most inflammatory markers or FAS. In summary, amongst the n3-PUFA, DHA was the most effective for elevating hepatic DHA levels, and preventing progression of hepatic steatosis via reductions in FAS and a marker of fibrosis.

## 1. Introduction

Non-alcoholic fatty liver disease (NFALD) represents a spectrum of disease ranging from steatosis (accumulation of intrahepatic fat) to non-alcoholic steatohepatitis [1]. NAFLD is highly associated with obesity and insulin resistance, given that 51% of individuals with obesity and up to 79% of patients with type 2 diabetes have NAFLD [2,3]. In obesity, excess calories are stored primarily in the visceral fat depots as triacylglycerides (TG), but then spill over for ectopic storage, mainly in the liver, and this progressively leads to hepatic steatosis. Furthermore, insulin resistance in obesity and type 2 diabetes results in less inhibition of lipolysis and less stimulation of lipoprotein lipase, which increases circulating free fatty acids and TG, thus providing more substrate for hepatic TG synthesis and storage [1]. 

Patients with hepatic steatosis have lower relative concentrations of n3-PUFA in the blood and in liver tissue biopsies (reviewed by the authors in reference [4]). This has led to an interest in whether supplementation of n3-PUFAs can reduce hepatic steatosis and delay the progression of NAFLD (reviewed by the authors in reference [5]). The results of some, but not all, n3-PUFA supplementation trials in humans have shown promise, particularly if docosahexaenoic acid (DHA, C22:6 n3) is increased in the liver (reviewed by the authors in reference [6]). N3-PUFAs include eicosapentaenoic acid (EPA, C20:5 n3) and DHA, which are present in marine sources and algae (reviewed by authors in reference [7]) and the plant-based dietary essential fatty acid α-linoleic acid (ALA, C18:3 n3), which can undergo elongation, desaturation, and oxidation to EPA and DHA. In animal models of hepatic steatosis induced by high-fat high-cholesterol diets, comparisons of EPA versus DHA supplementation show that both fatty acids reduce hepatic steatosis, although there are some differential effects on specific parameters such as liver lipid levels, inflammation, and fibrosis [8,9,10]). Dietary interventions with ALA-rich oils such as flaxseed oil, perilla oil, or *Echium* oil also reduce hepatic steatosis, inflammatory biomarkers, fibrosis, and oxidative stress in animal models using high-fat diets with or without cholesterol to induce hepatic steatosis [9,11,12,13]. ALA, EPA, and DHA supplementation have been compared in one study using a rodent model of high-carbohydrate high-fat diet-induced metabolic syndrome characteristics and it was reported that each of the n3-PUFAs was effective for reducing hepatic steatosis and inflammation [14]. However, the authors noted that EPA and DHA were more effective in the control groups receiving low-fat diet compared to the metabolic syndrome groups receiving the high-carbohydrate high-fat diet, suggesting that it is the proportion of fatty acids in the dietary lipid pool, versus the diet as a whole, that is most important for determining n3-PUFA responses [14]. Thus, an important limitation of the published studies with animal models is that n3-PUFA supplementation is studied in the context of high-fat diets, whereas the only current effective strategy for treating hepatic steatosis in humans (those with obesity or type 2 diabetes; adults and adolescents) is lifestyle intervention involving reduced caloric intake and exercise [1]. Since it is still unclear which of the n3-PUFAs is effective in the early stages of hepatic steatosis and whether the protective effects of n3-PUFA supplementation can be achieved with low-fat diets, the present study employed *fa/fa* Zucker rats as the model since they develop obesity, insulin resistance, and hepatic steatosis on low-fat diets (<10% *w*/*w* or <25% calories) that are responsive to various dietary interventions [15,16]. Thus, the overall objective of this study was to directly compare the n3-PUFAs, plant-based ALA in flaxseed oil, and marine-based EPA or DHA in high-purity oils, for their effects on hepatic steatosis, markers of hepatic inflammation and fibrosis, and insulinemia in *fa/fa* Zucker rats. We also investigated whether the underlying mechanisms involved changes in fatty acid synthesis or oxidation, and/or insulin signalling. The results revealed that dietary DHA and EPA operate by different mechanisms to modulate hepatic steatosis and hyperinsulinemia in *fa/fa* Zucker rats, and that DHA was the most effective among the n3-PUFA for elevating hepatic DHA levels, preventing progression of hepatic steatosis via reduced FAS, and reversing a marker associated with fibrosis despite elevation of some indicators of inflammation.

## 2. Materials and Methods

### 2.1. Experimental Design

Five week old male *fa/fa* Zucker and lean (*+/?*) Zucker rats (Charles River Laboratories, St-Constant, PQ) underwent a minimum 1 week acclimation period and were fed a diet based on the AIN-93G diet containing soybean oil [17]. The *fa/fa* Zucker rats were randomly assigned (*n* = 10 rats/group) to the baseline group (faBASE; tissue collections at the end of the acclimation period) or to an 8 week intervention with diets containing n3-PUFA from ALA (faALA), EPA (faEPA), or DHA (faDHA), or n6-PUFA from LA (faLA). The lean Zucker rats served as a healthy reference group and were fed a diet containing n6-PUFA from LA (lnLA) for 8 weeks. The diet formulations and the fatty acid composition of the diets (as analyzed by gas chromatography) are shown in Table 1. The diets contained 10% (*w*/*w*) total fat and were formulated with oil mixtures to keep the saturated (SFA), monounsaturated (MUFA), and polyunsaturated fatty acid (PUFA) content consistent amongst the diets. The dose of 3% (*w*/*w*) EPA or 3% (*w*/*w*) DHA was chosen to avoid potential complications associated with higher doses [18] and was achieved using purified EPA or DHA oil (>95% purity and in free fatty acid form). ALA was provided at the same dose (3% *w*/*w*) using flaxseed oil as the source of ALA (in triglyceride form). Rats were singly caged and provided free access to the diets. Feed intake corrected for spillage, and body weights were recorded for all groups. All animal care procedures were approved by the University of Manitoba Animal Care Committee (Protocol 12-050) and conducted according to guidelines of the Canadian Council on Animal Care. 

At the end of the acclimation period (faBASE) or 8 week dietary intervention (experimental groups), the rats were fasted overnight, and euthanized by carbon dioxide asphyxiation, followed by cervical dislocation. Trunk blood was collected. The liver was weighed and portions were immediately frozen in liquid nitrogen and stored at −80 °C or embedded in Optimal Cutting Temperature (OCT) compound and frozen in a dry ice-ethanol bath. 

### 2.2. Serum Biochemistry

Fasting serum was analyzed for glucose (Genzyme Diagnostics P.E.I. Inc., Charlottetown, PE, Canada), insulin (MesoScaleDesign, Gaithersburg, MD, USA), and haptoglobin (Tri-Delta Diagnostics, Inc., Maynooth, Ireland) using commercial kits. The homeostasis model assessment-insulin resistance (HOMA-IR) index was calculated as fasting serum insulin (µU/Ml) × fasting plasma glucose (mmol/L)/22.5 [19]. A Cobas c111 clinical chemistry analyzer (Roche Diagnostics, Indianapolis, IN) was used to quantify serum total cholesterol, HDL-cholesterol, LDL-cholesterol, TG, alanine aminotransferase (ALT), and aspartate aminotransferase (AST). 

### 2.3. Hepatic Lipid and Fatty Acid Composition

Total lipid was extracted from liver tissue using chloroform/methanol and quantified gravimetrically as previously described [20]. Total lipid in diet samples was also extracted with chloroform/methanol. The chloroform/methanol extract from liver tissue was separated into TG and PL fractions via thin layer chromatography as previously described in reference [21]. The fatty acids in the hepatic TG and PL fractions, or diet samples, were methylated with methanolic HCl and run on a Varian 450-GC Gas Chromatograph (Varian, Mississauga, ON, Canada) using a 100 m × 0.25 mm diameter and 0.25 μm film thickness GC capillary column (Varian, Mississauga, ON, Canada). Individual fatty acids were identified by retention time based on fatty acid standards.

### 2.4. Hepatic Lipid Droplets

Liver frozen in OCT compound was sectioned (5 μm) and stained with Oil Red O to visualize hepatic lipid droplets [15]. Sections were visualized with a Zeiss Axioskop 2 plus microscope (Zeiss, Thornwood, NY, USA) and images were captured with a Zeiss Axiocam digital camera using Axio Vision 4.6 (Zeiss, Thornwood, NY, USA). Quantification of lipid droplet number and size was carried out using ImageJ software (National Institute of Health, Bethesda, MD, USA) [22]. For lipid droplet size measurements, 25 adjacent lipid droplets were randomly selected from 4 different sections of liver tissue for a total of 100 lipid droplets for each rat. For quantification of lipid droplet number, liver sections were divided into 0.01 mm^2^ squares and all lipid droplets within this area were counted. 

### 2.5. Western Immunoblotting

Protein extract preparation, sodium dodecyl sulfate-polyacrylamide gel electrophoresis (SDS-PAGE), protein transfer to polyvinylidene difluoride membrane and antibody labelling were completed as previously described [23]. The membranes were immunoblotted with primary antibodies to sterol regulatory element-binding protein-1c (SREBP-1c) (C-20), Acetyl CoA Oxidase (ACO) (Santa Cruz Biotechnology, Santa Cruz, CA), fatty acid synthase, FoxO1, phospho-Akt (p-Akt; Ser473), phospho-AMP kinase-α (p-AMPK-α; Thr172), phospho-NF-κB p65 (Ser536), α-smooth muscle actin (Cell Signaling Technology, Danvers, MA), or F4/80 (Abcam, San Francisco, CA), with β-tubulin (Cell Signaling Technology, Danvers, MA) or Ponceau staining used to control for protein loading. The HRP-conjugated secondary antibody was used to detect the protein bands, which were imaged by scanning densitometry with a model GS-800 Imaging Densitometer (Bio-Rad Laboratories, Hercules, CA) after visualization with chemiluminescence reagent (Amersham^TM^ ECL^TM^ Prime Western Blotting Detection Reagent). The bands of interest were quantified using AlphaView SA software (Protein Simple, Santa Clara, CA), and their intensities were expressed relative to the intensity of suitable loading control.

### 2.6. Statistical Methods

Data were analyzed by a one-way ANOVA (SAS Version 9.2, SAS institute, Cary, NC, USA) followed by Duncan’s multiple range test for post-hoc testing. Data that were not normal or homogeneous after log transformation were analyzed by non–parametric testing with the Kruskal-Wallis test and then least significant differences with Tukey correction for multiple comparisons. The Chi-squared test was used to analyze the liver lipid droplet size distribution. Outliers (± 2.5 standard deviations from the mean) were removed from the dataset before analysis. The data are reported as means ± the standard error of the mean (SEM) and differences were considered significant at *p* < 0.05.

## 3. Results

### 3.1. Feed Intake and Body Weight

The *fa/fa* Zucker rats had a greater initial body weight than the lean Zucker rats, with no differences in initial body weight among the *fa/fa* Zucker, which were fed the experimental diets for eight weeks (Table 2). The plant oil fed groups (both faLA and faALA) had a greater feed intake and higher final body weight than the marine oil fed groups (faEPA and faDHA). During the eight week study, the *fa/fa* rats had a lower feed efficiency ratio than the lnLA group, however, the feed efficiency ratio was not different among the *fa/fa* groups. 

### 3.2. Hepatic Steatosis

Absolute liver weight was greater in the faALA group compared to the faLA, faEPA, and faDHA groups, and 2-fold higher in the 14 week old *fa/fa* rats compared to the lnLA group (Table 2). Similarly, the liver weight as a percent body weight was 1.7-fold greater in *fa/fa rats* than in the lnLA group, however, liver weight expressed relative to body weight did not differ among the *fa/fa* groups. The faLA and faALA groups had the highest concentration of liver lipid and a level that was 3-fold greater than the lnLA group (Figure 1A). The faEPA and faDHA groups had 24% and 36% lower hepatic lipid concentrations, respectively, compared to the faALA group. The hepatic lipid concentration of the faEPA and faDHA groups was not different from the faBASE group, suggesting that marine-based n3-PUFA prevented accumulation of lipid in the liver during the eight week study. 

Liver sections were stained with Oil Red O to visualize lipid droplets (Appendix A). Hepatic lipid droplet assessment showed no differences in the average size of lipid droplets among the *fa/fa* experimental groups (Figure 1B) but they were all significantly larger than the faBASE and lnLA groups, which were too small in size to quantify by the procedure employed in the present study. With respect to the size distribution of the lipid droplets, Chi-squared testing confirmed that there were differences among the dietary groups for all the size ranges. The faLA group had 14% of lipid droplets in the largest size range (>300 µm^2^) compared to only 1–6% for the other groups (Figure 1C). The faEPA and faDHA groups had 5% of lipid droplets in the 201–300 µm^2^ range compared to 12–13% for the faLA and faALA groups. In contrast, the faDHA group had 81% of its lipid droplets in the smallest size range (0–100 µm^2^) compared to 62% for faEPA, 47% for faALA, and 52% for faLA.

Since the differences in hepatic steatosis among groups could be due to an imbalance between fatty acid synthesis and oxidation, Western immunoblotting was used to assess key proteins in the regulation of fatty acid metabolism (SREBP-1c), synthesis (FAS) and oxidation (ACO). There were no differences seen in hepatic protein levels of SREBP-1c or activated SREBP-1c (68 kDa/(68 kDa + 125 kDa)), or ACO (data not shown). However, hepatic protein levels of FAS were significantly reduced 40–52% in the faEPA group compared to the faLA, faALA, and faBASE groups (Figure 1D). FAS levels were reduced further (by 45%) in the faDHA group compared to the faEPA group, reaching levels not different from the lnLA group. 

### 3.3. Biomarkers Related to Hepatic Function

ALT and AST are normally intracellular enzymes, but serum levels rise when the hepatic function is compromised. Serum levels of ALT and AST were elevated in the faALA group compared to all the other groups (Table 2). Serum ALT and AST for the faEPA and faDHA groups were not different from faBASE or lnLA. In contrast, the faEPA and faDHA groups had elevated concentrations of serum haptoglobin, an acute phase protein, compared to faLA and faALA groups, and all these groups had higher haptoglobin concentrations than the faBASE group. F4/80, a marker of macrophage infiltration, was elevated 2.6–2.9 fold in the faALA and faBASE groups compared tolnLA, while the faLA, faEPA and faDHA groups had intermediate levels (Figure 2A). In contrast, phospho-NF-κB p65 (Ser536) levels were 20-fold greater in the faEPA and faDHA groups compared to the faBASE group, while faALA was 6-fold higher than the faBASE rats (Figure 2B). Furthermore, phospho-NF-κB p65 (Ser536) was not different among lnLA, faLA, and faALA groups. Interestingly, the lnLA, faLA, faEPA groups had a ~55% reduction in levels of α-smooth muscle actin (α-SMA) compared to the faBASE group, and this marker of fibrosis was further reduced by 67% in the faDHA group (Figure 2C).

Accumulation of lipid in the liver is associated with insulin resistance. Although fasting glucose was unchanged among *fa/fa* groups, fasting serum insulin was 40–45% lower in the faLA and faDHA compared to the faALA and faEPA groups, respectively, but none were as low as the lnLA or the faBASE groups (Figure 3A,B). On the other hand, the faEPA group had a higher HOMA-IR score, an indication of insulin resistance, than the faLA group, whereas the faALA and faDHA groups had values that were intermediate and not significantly different from either faEPA or faLA groups (Figure 3C). The HOMA-IR score for the faLA group was still 5-fold higher than the faBASE group. We examined hepatic levels of FoxO1, a mediator of the insulin signaling pathway that regulates gluconeogenesis. The lnLA and faEPA groups had reduced levels of FoxO1 compared to all other groups (Figure 3D), suggesting greater inhibition of hepatic gluconeogenesis and thus greater responsiveness to insulin. pAkt (Ser473), a mediator of insulin signaling, was elevated 2- to 3-fold in lnLA rats compared to the *fa/fa* groups, which showed no differences due to dietary n3-PUFA or n6-PUFA (Figure 3E). p-AMPK-α (Thr172), a cellular energy sensor, was not different among the groups (Figure 3F).

With respect to circulating lipid levels (Table 2), the faLA and faALA groups had the highest total cholesterol, while all the other experimental groups had similar total cholesterol concentrations. LDL-C was highest in the faLA group, followed by the faALA group, and both were significantly higher than the other groups. The faLA group also had the highest concentration of HDL-C followed by the faALA group, and these were significantly higher than the other groups. The faDHA group had the lowest HDL-C concentration among the groups. The faEPA had lower serum TG concentrations than the faBASE but not the lnLA group. 

### 3.4. Hepatic TG and PL Fatty Acid Composition

A summary of fatty acid classes and fatty acid ratios for hepatic TG and PL is shown in Table 3, whereas individual fatty acids of interest in hepatic TG and PL are highlighted in Figure 4 and Figure 5, respectively. The lnLA group had the lowest proportion of SFA in TG but it was similar to the faDHA group. The faALA group had the highest total MUFA in TG among the fa/fa experimental groups. The lnLA had the highest total PUFA in TG, while the faLA and faALA groups had the least. The faDHA had the highest total n3-PUFA in hepatic TG while faALA and faEPA groups had 2-fold less. The faLA, faEPA, and faDHA groups had 2-fold more n6-PUFA in TG compared to the faALA group. However, the faLA group had an n6/n3 ratio of 64 in TG, while this ratio was 1 to 2 in the faALA, faEPA, and faDHA groups. When lean and fa/fa rats were fed the same LA diet, the obese rats had more SFA and MUFA but less total n3, and n6-PUFA in TG and a higher n6/n3 ratio compared to their lean counterparts. In contrast, there were no differences among the groups with respect to SFA or MUFA in the PL fraction. The faDHA group had a higher proportion of PUFA in PL compared to the faLA group. Total n3-PUFA in PL was not different among the faALA, faEPA and faDHA groups. The fa/fa experimental rats had similar n6-PUFA in PL. The faLA group had an n6/n3 ratio of 4 in PL whereas this ratio was 1 to 2 for the faALA, faEPA, and faDHA groups. When fed the LA diet, fa/fa rats had less total and n6-PUFA but more n3-PUFA than lean rats.

With respect to individual fatty acids, the n3 fatty acid profile of the diets was reflected in the n3 fatty acid profile of hepatic TG and PL (Figure 4 and Figure 5). The faALA group had the highest ALA (Figure 4A and Figure 5A), the faEPA group had the highest EPA and DPA (Figure 4B,C and Figure 5B,C), and the faDHA group had the highest DHA (Figure 4D and Figure 5D), with the greatest magnitude difference in the TG fraction compared to the PL fraction. The faALA and faDHA groups had similar proportions of DPA in TG and EPA in PL, whereas the faALA group had less EPA in PL and more DPA in PL than the faDHA group. Although the faDHA group had 2-fold higher DHA in PL compared to the faALA group, the faALA group had 74% and 22% more DHA in PL compared to the faEPA and faLA groups, respectively. The only n3-PUFA difference between lnLA and faLA groups was elevated DHA in PL of faLA compared to lnLA. 

With respect to n6 fatty acids, the faLA group had higher LA in TG than faALA and faEPA groups but not the faDHA group (Figure 4E). The faLA group had higher γ-linolenic acid (GLA) and arachidonic acid (AA) in TG than the faALA, faEPA, and faDHA groups, and higher dihommo-gamma-linolenic acid (DGLA) in TG than the faALA group (Figure 4F–H). The faALA and faDHA groups had a higher proportion of LA and DGLA in the PL fraction than the faLA and faEPA groups (Figure 5E,G). However, the faLA group had a higher proportion of GLA in PL compared to the faALA and faEPA groups, and a higher proportion of AA in PL compared to all three n3 *fa/fa* groups (Figure 5F,H). The faDHA group had less AA in PL than the faALA group while the faEPA group was intermediate. The lnLA rats had a higher proportion of all n6-PUFAs in TG and PL compared to faLA except for DGLA and AA in PL.

## 4. Discussion

The major finding of the present study is that diets containing DHA or EPA prevented the progression of hepatic steatosis in *fa/fa* Zucker rats as demonstrated by total lipid concentrations not different from baseline, however, DHA and EPA evoked different mechanisms related to fatty acid metabolism, inflammation, and insulinemia. DHA was more effective than EPA for reducing hepatic levels of FAS, increasing the proportion of smaller lipid droplets in the liver, attenuating fasting hyperinsulinemia, and ameliorating insulin resistance (HOMA-IR) in *fa/fa* Zucker rats. The reduction in hepatic steatosis in DHA-fed *fa/fa* Zucker rats was associated with the lowest levels of a fibrosis marker (α-SMA); furthermore, the 5-fold reduction of α-SMA from baseline by DHA indicates that reversal of early fibrotic damage is possible. The faDHA rats had the highest proportion of DHA in liver PL and TG fractions, supporting the view that higher tissue levels of DHA are protective with respect to hepatic steatosis. However, it is plausible that DHA, and to some extent EPA, may be directly regulating fatty acid synthesis, as demonstrated by reduced FAS levels, and thus prevent hepatic lipid accumulation through this mechanism. Dietary DHA and EPA reduced AA in hepatic TG and PL to a similar extent, and this would reduce the availability of AA for group IVA phospholipase A_2_ and the downstream AA cascade which has been implicated in the development of hepatic steatosis [24]. EPA, but not the other n3-PUFAs, reduced hepatic FoxO1 which suggests greater inhibition of gluconeogenesis; however, this potential mechanism did not explain the improvements in fasting hyperinsulinemia, or the HOMA-IR score observed in the faEPA group. In contrast, dietary ALA, a plant-based n3-PUFA, did not attenuate hepatic steatosis despite having similar DHA and AA in liver PL as compared to the faEPA group. Hepatic FAS levels were not altered in the faALA group compared to baseline, and markers of inflammation (haptoglobin, F4/80) and hepatic function (ALT, AST) remained elevated. On the other hand, the faEPA and faDHA groups had 3-fold higher hepatic p-NF-κB p65 protein and elevated circulating haptoglobin compared to faALA, suggesting that n3-PUFA differentially regulates the various inflammatory pathways. Although n6-PUFA did not alter hepatic steatosis and elevated circulating cholesterol (total, LDL-C and, HDL-C), the faLA group had attenuated fasting insulinemia and insulin resistance (HOMA-IR) similar to the faDHA group. This indicates that improvements in insulin-related parameters due to n6-PUFA consumption can occur without changes in liver lipid content or markers of fatty acid metabolism. Furthermore, consumption of n6-PUFA did not have a pro-inflammatory effect in *fa/fa* Zucker rats in the context of elevated hepatic lipid as markers of inflammation (F4/80, p-NF-κB p65) and hepatic function (ALT, AST) were not increased in the faLA group relative to baseline Zucker rats.

The present study demonstrates that both DHA and EPA prevent the accumulation of liver lipid relative to baseline levels over eight weeks, but that DHA is more effective than EPA for reducing FAS levels, promoting smaller lipid droplets and improving insulin resistance in the *fa/fa* rodent model of genetic obesity, insulin resistance, and hepatic steatosis. Likewise, an inhibitor of mammalian FAS has been shown to reduce hepatic lipids and improve insulin sensitivity in various preclinical models [25]. The parallel changes in absolute liver weight and liver lipid content support the evidence that DHA and EPA reduced hepatic lipid deposition. Cellular energy state was not a factor in the present study as hepatic levels of pAMPK were not different among the groups. With respect to dietary n3-PUFAs, Depner et al [8] reported that diets containing 2% energy from EPA, DHA, or a combination of EPA and DHA for 16 weeks did not prevent hepatosteatosis when induced by a Western-type high-fat high-cholesterol diet in *Ldlr^-/-^* mice. On the other hand, Suzuki-Kemuriyama et al [10] reported that 5% EPA was more effective than 5% DHA for reducing hepatic triglycerides when C57/BL6 mice were fed a high-fat high-sucrose atherogenic diet containing 1.25% cholesterol and 0.5% cholate, however, this was only a four week study. Likewise, Poudyal et al [14] reported 50% lower liver lipid concentrations in rats consuming EPA versus DHA (final dose not specified) for eight weeks after induction of metabolic syndrome characteristic during the previous eight weeks with a high-carbohydrate high-fat diet plus 25% fructose in the drinking water. Thus, it appears that the effects of marine n3-PUFA are dependent on the model and the dose. The present study differs with respect to the species and genetic model (*fa/fa* rats with a mutation in the leptin receptor) as well as the diet, since hepatic steatosis in *fa/fa* rats occurs with consumption of lower fat diets (10% w/w or 23% energy from fat) and without diet ingredients such as cholesterol, cholate, or fructose. In addition, the dose of n3-PUFA (3% of energy) in the present study represents a greater proportion of the dietary fat than in the aforementioned studies [8,10,14] and perhaps this is a contributing factor to the efficacy of DHA or EPA preventing further development of hepatic steatosis in growing *fa/fa* rats.

In the present study, an ALA-rich diet did not reduce hepatic steatosis despite having DHA, AA, and the n6/n3 ratio in PL similar to the faEPA group. Although there has been considerable focus on the relationship between hepatic DHA levels and NAFLD [4], limiting AA may also be important as IVA phospholipase A_2_ plays a role in hepatic lipid deposition, macrophage filtration, and progression of hepatic fibrosis [24]. We have previously reported that a diet high in ALA and MUFA attenuated hepatic steatosis in diet-induced obese rats and was associated with high levels of EPA and DHA and low levels of AA in hepatic PL [11]. Botelho et al [9] found that *Echium* oil (ALA + stearidonic acid) and fish oil (EPA + DHA) but not algal oil (DHA) reduced hepatic steatosis (assessed histologically) when *Ldlr^-/-^* mice fed high fat diets were provided the oils by gavage for four weeks; the attenuation in hepatic steatosis with the *Echium* oil was attributed to a reduced n6/n3 ratio in liver. In the study by Poudyal et al [14], liver lipid accumulation was reduced 75% by EPA supplementation, 57% by the ALA-rich chia oil, and 41% by DHA supplementation compared to the control diet when Wistar rats consumed high-carbohydrate high-fat diets and drinking water with 25% fructose for eight weeks followed by treatment for eight weeks with the oils containing n3-PUFAs. Supplementing a high-fat high-cholesterol diet with 5.5% (w/w) perilla oil as a source of ALA for 16 weeks reduced the hepatic steatosis score in Wistar rats and was associated with greater fecal cholesterol and bile acid secretion in this steatosis model with high cholesterol intake [12]. More recently, Han et al [13] reported that consumption of a Western-type high-fat high-cholesterol diet containing 10% (w/w) flaxseed oil (high ALA) for 12 weeks substantially reduced hepatic fat accumulation in *ApoE*^-/-^ mice. 

When studies simply substitute perilla oil or flaxseed oil for lard in the high fat diet to increase n3-PUFA from ALA, this also changes the proportions of SFA, MUFA, and PUFA in the diet and confounds the interpretation of the results as it is unclear whether the changes are due to ALA, a reduction in n6-PUFA, or an increase in the dietary PUFA/SFA ratio. The current study was designed to change the type of n3-PUFA while keeping the proportions of SFA, MUFA, and PUFA similar among the diets; perhaps this is a contributing factor as to why there was no improvement in hepatic steatosis in the faALA group. We have previously reported that a diet high in ALA and MUFA attenuated hepatic steatosis in diet-induced obese rats when a canola/flaxseed oil mixture replaced lard in the high-fat diet [11] and the diets contained 8:54:18:20% or 49:42:8:1%, respectively, for the proportions of SFA:MUFA:n6-PUFA:n3-PUFA. Furthermore, the current study compared ALA, EPA, and DHA at the same dose, but this may not be the optimal dose for any of these n3-PUFAs. Higher doses of purified EPA or DHA may have unintended consequences [18], while employing them at high doses for management of a disease condition such as NAFLD becomes a pharmaceutical approach. Some of the human trials to date indicate that n3-PUFA supplementation with EPA and/or DHA may be beneficial for reducing liver fat content but not histological measures of NAFLD or fibrosis [6,26].

DHA was superior to EPA for reversing early stage fibrosis in the liver as indicated by lower α-SMA levels compared to baseline *fa/fa* rats. Interestingly, α-SMA was lowest in the faDHA group, being 50% lower than in lnLA, whereas faEPA and faLA groups had α-SMA levels similar to lnLA. The elevated hepatic protein levels of α-SMA in the faBASE group suggest that the diet provided in the early weeks of life was promoting fibrosis. However, it needs to be noted that others have reported that *fa/fa* Zucker rats are resistant to fibrosis induced chemically or with a Western diet (based on mRNA levels of α-SMA and histological assessment) as leptin receptor-mediated signaling appears to play a role in development of hepatic fibrosis and remodeling of the extracellular matrix [27,28]. Further histological assessment and additional fibrotic markers are required for a more in-depth understanding of the reduction in hepatic α-SMA with DHA intervention in the current study. With respect to n3-PUFA intervention, others have reported that DHA consumption resulted in greater suppression of hepatic fibrosis markers at the mRNA level compared to EPA in *Ldlr^-/-^* mice fed a Western-type diet [8], whereas Suzuki-Kemuriyama et al [10] found that both DHA and EPA decreased hepatic fibrosis assessed by α-SMA protein levels in mice fed an atherogenic high-fat diet. In the present study, the higher α-SMA levels of the faALA group paralleled the degree of hepatic steatosis. On the other hand, Chen et al [12] found that ALA-rich perilla oil reduced hepatic steatosis and inhibited hepatic fibrosis based on histological assessment in rats fed a high-fat high-cholesterol diet suggesting that the model for induction of hepatic steatosis may be a factor in the level of fibrosis and responsiveness to dietary intervention.

Enzymes such as ALT and AST were used to evaluate liver function, and are present in higher levels in the circulation only if released from hepatocytes via damage to the cell membranes. Given that DHA and EPA prevented the progression of hepatic steatosis, it is not surprising that these two groups had serum ALT and AST concentrations that were not different from faBASE. On the other hand, the faALA group had serum ALT and AST levels that were elevated 2-fold concurrently with hepatic steatosis. Others have reported that reduced levels of serum ALT and AST accompany attenuation of liver lipids due to dietary interventions with ALA, EPA, or DHA in various models [10,13,14]. 

The anti-inflammatory properties of n3-PUFAs are postulated to reduce the hepatic inflammation associated with hepatic steatosis [6]. However, in the present study, n6-PUFA did not worsen the initial inflammatory state of genetically obese rats as markers of inflammation (F4/80, p-NF-κB p65) and hepatic function (ALT, AST) were not increased in the faLA group relative to baseline *fa/fa* Zucker rats. In contrast, liver lipid accumulation in the faALA group was associated with macrophage infiltration (↑F4/80) and necrosis (↑ALT, ↑AST) while reductions in liver lipid content in the faEPA and faDHA groups were associated with both systemic inflammation (↑haptoglobin) and hepatic inflammation (↑ p-NF-κB p65). This is an important example showing that n3-PUFAs can have differential effects on inflammatory pathways and that attenuation of hepatic steatosis does not necessarily resolve inflammation. Others have used a variety of approaches to assess hepatic inflammation and have concluded that DHA is more effective than EPA based on greater suppression of various inflammatory markers at the mRNA level [8], histological assessment of inflammatory cell infiltration, immunohistochemical assessment of F4/80, immunoblotting for c-Jun N-terminal kinase JNK activation, and hepatic mRNA levels of tumor necrosis factor-α (TNF-α) and monocyte chemoattractant protein-1 (MCP-1) [10]. Based on histology, Poudyal et al [14] reported that dietary ALA, EPA, and DHA supplementation diminished portal inflammation, and Chen et al [12] reported that ALA-rich perilla oil reduced the area for inflammatory cell infiltration. Han et al [13] found that ALA-rich flaxseed oil decreased hepatic protein levels of interleukin-6 (IL-6), TNF-α and MCP-1, as well as phosphorylation of the NF-κB p65 subunit. However, it is noteworthy that reductions in hepatic inflammation do not necessarily reflect levels of systemic inflammation. In the present study, serum haptoglobin, an acute phase protein and non-specific marker of systemic inflammation that is present in higher amounts in rats compared to C-reactive protein [29], was elevated in the faEPA and faDHA groups despite having hepatic lipid levels similar to baseline. On the other hand, Botelho et al [9] reported that four-week supplementation of oils rich in ALA, EPA, or DHA did not alter circulating biomarkers of inflammation such as CRP or IL-6 in *LDLr*^-/-^ mice, whereas Han et al [13] found that flaxseed oil supplementation for 12 weeks decreased plasma IL-6, TNF-α, and MCP-1 in Apo E^-/-^ mice. Emerging clinical trial results, such as COMPASS (Cardiovascular Outcomes for People Using Anticoagulation Strategies) that investigated Rivaroxaban, an inhibitor of the IL-1β pathway, are indicating that pharmaceuticals targeting specific inflammatory pathways are capable of reducing inflammation and cardiovascular events [30], and thus pathway specificity needs to be addressed in future studies investigating dietary n3-PUFA and their potential anti-inflammatory effects in various disease conditions.

Interestingly, the faLA group had an improvement in HOMA-IR due to a small reduction in fasting hyperinsulinemia, however, this occurred despite no improvement in hepatic steatosis or insulin signaling based on hepatic pAkt. On the other hand, both DHA and EPA prevented significant liver lipid accumulation compared to the baseline group, but the faDHA group had 37% lower fasting hyperinsulinemia compared to the faEPA group. Likewise, dietary EPA and DHA had differing effects on FoxO1 in the liver of *fa/fa* rats. In fact, the lower levels of FoxO1 in the faEPA group, similar to the lean Zucker rats, would suggest an improvement in insulin signaling and greater inhibition of hepatic gluconeogenesis, however, this did not translate to a reduction in their fasting hyperinsulinemia or improvement in their HOMA-IR score. Furthermore, a novel finding of the current study is that FoxO1 is not mediating the beneficial effects of DHA or LA on insulinemia. The functioning of FoxO1 requires phosphorylation and acetylation [31], and perhaps there is also a limitation for phosphorylation or acetylation in the *fa/fa* rats that is not being overcome with DHA supplementation or a diet enriched in LA. 

## 5. Conclusions

In summary, the DHA diet was more effective than EPA for preventing progression of hepatic steatosis in *fa/fa* obese rats as demonstrated by less hepatic lipid accumulation and more of the smaller lipid droplets, as well as the associated reduction in hyperinsulinemia. At a cellular level, the elevated DHA and reduced AA in hepatic TG and PL may have contributed directly or indirectly to greater inhibition of fatty acid synthesis, resulting in reduced lipid deposition, improved insulin sensitivity, and less fibrosis. In addition to lower levels of FAS, the EPA diet resulted in a novel reduction of FoxO1, although this did not alter the hyperinsulinemia. Based on the divergent responses of the various inflammatory markers, future studies need to address pathway specificity when investigating the anti-/pro-inflammatory effects of dietary n3-PUFA and n6-PUFA in various disease conditions. Further studies are needed to define the underlying mechanism(s) for the differential effects of DHA and EPA on pathways contributing to the pathogenesis of NAFLD in order to develop better management strategies.

## Figures and Tables

**Figure 1 nutrients-11-00917-f001:**
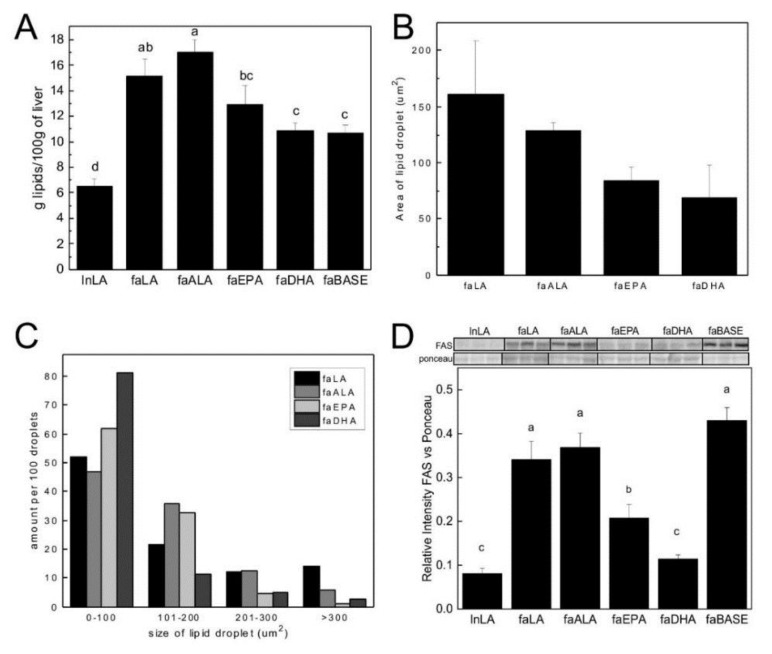
Lipid concentration (**A**), average lipid droplet size (**B**), lipid droplet size distribution (**C**), and fatty acid synthase (FAS) protein levels (**D**) in the liver. The relative levels of FAS (**D**) in the liver were obtained using Western blotting. Representative bands are shown in the upper panel, with individual bands arranged in the same order as the data in the graphs. All bands were taken from the same blot and exposure without any other manipulation. Densitometry was used to quantify the intensity of the bands. Data were normalized to a band identified by Ponceau staining. Values are expressed as mean ± SEM for *n* = 6–7/group (**A**), *n* = 4/group (**B**,**C**) and *n* = 5–6/group (**D**); different letters (a,b,c,d) indicate significant differences (*p* < 0.05) among the means, and an absence of letters indicates no significant differences. Values for the lipid droplet size distribution (C) are expressed as number per 100 droplets; Chi-squared testing confirmed that there were differences among the dietary groups for all the size ranges. Abbreviations: faALA, *fa/fa* Zucker rats fed the α-linolenic acid diet; faBASE, baseline *fa/fa* Zucker rats; faDHA, *fa/fa* Zucker rats fed the docosahexaenoic acid diet; faEPA, *fa/fa* Zucker rats fed the eicosahexaenoic acid diet; faLA, *fa/fa* Zucker rats fed the linoleic acid diet; FAS, fatty acid synthase; lnLA, lean Zucker rats fed the linoleic acid diet.

**Figure 2 nutrients-11-00917-f002:**
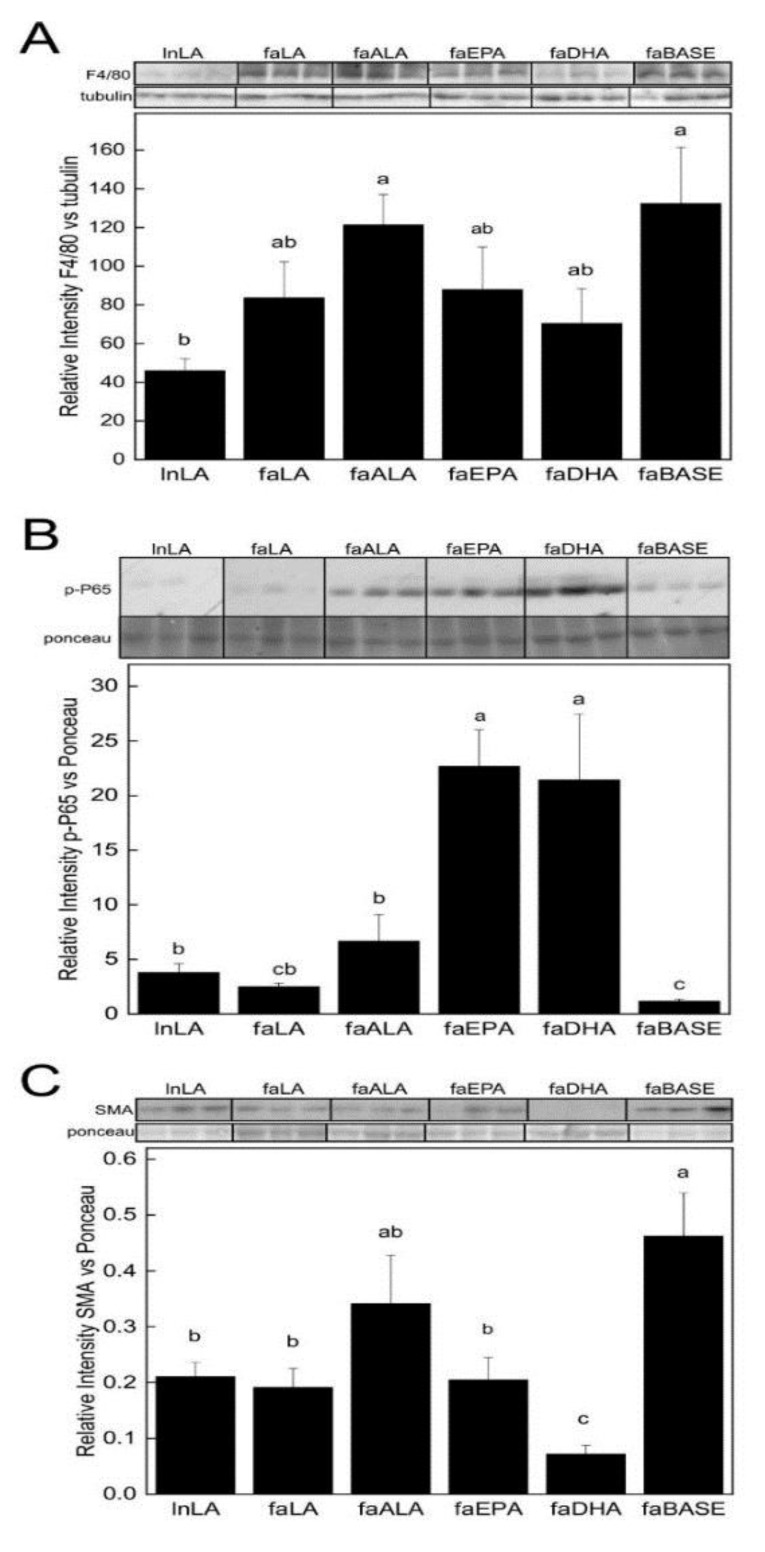
Hepatic protein levels of F4/80 (**A**), p-NF-κB p65 (B) and α-SMA (**C**). The relative levels of F4/80, a marker of macrophage infiltration (**A**), p-NF-κB p65, a marker of inflammation (**B**), and (**C**) α-SMA, a fibrosis marker, in the liver were obtained using Western blotting. Representative bands are shown in the upper panel, with individual bands arranged in the same order as the data in the graphs. All bands were taken from the same blot and exposure without any other manipulation. Densitometry was used to quantify the intensity of the bands. Data were normalized to β-tubulin or a band identified by Ponceau staining. The results are presented as means ± SEM; *n* = 5–6/group. Statistical differences (*p* ≤ 0.05) among means are indicated by different lower case letters (a,b,c). Abbreviations: faALA, *fa/fa* Zucker rats fed the α-linolenic acid diet; faBASE, baseline *fa/fa* Zucker rats; faDHA, *fa/fa* Zucker rats fed the docosahexaenoic acid diet; faEPA, *fa/fa* Zucker rats fed the eicosahexaenoic acid diet; faLA, *fa/fa* Zucker rats fed the linoleic acid diet; lnLA, lean Zucker rats fed the linoleic acid diet; α-SMA, α-smooth muscle actin.

**Figure 3 nutrients-11-00917-f003:**
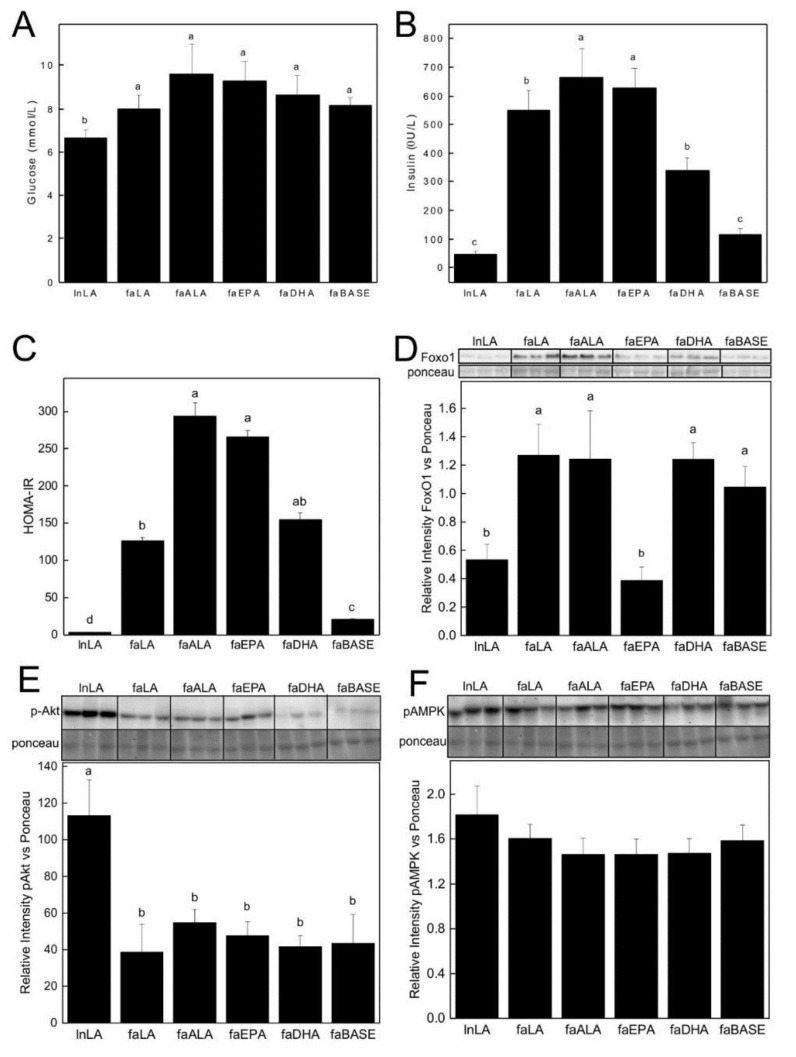
Fasting serum glucose (**A**) and insulin (**B**), HOMA-IR (**C**), and hepatic FoxO1 (**D**), pAkt (**E**) and pAMPK (**F**) protein levels. The relative levels of FoxO1 (**D**), pAkt (**E**) and pAMPK (**F**) in the liver were obtained using Western blotting. Representative bands are shown in the upper panel, with individual bands arranged in the same order as the data in the graphs. All bands were taken from the same blot and exposure without any other manipulation. Densitometry was used to quantify the intensity of the bands and data were normalized to a band identified by Ponceau staining. Values are expressed as means ± SEM with *n* = 6–10/group (**A**–**C**) or *n* = 5–6/group (**D**–**F**). Different superscript letters (a,b,c,d) indicate significant differences (*p* < 0.05) among the means. Abbreviations: faALA, *fa/fa* Zucker rats fed the α-linolenic acid diet; faBASE, baseline *fa/fa* Zucker rats; faDHA, *fa/fa* Zucker rats fed the docosahexaenoic acid diet; faEPA, *fa/fa* Zucker rats fed the eicosahexaenoic acid diet; faLA, *fa/fa* Zucker rats fed the linoleic acid diet; FoxO1, forkhead box protein O1; homeostasis model assessment-insulin resistance, HOMA-IR; lnLA, lean Zucker rats fed the linoleic acid diet.

**Figure 4 nutrients-11-00917-f004:**
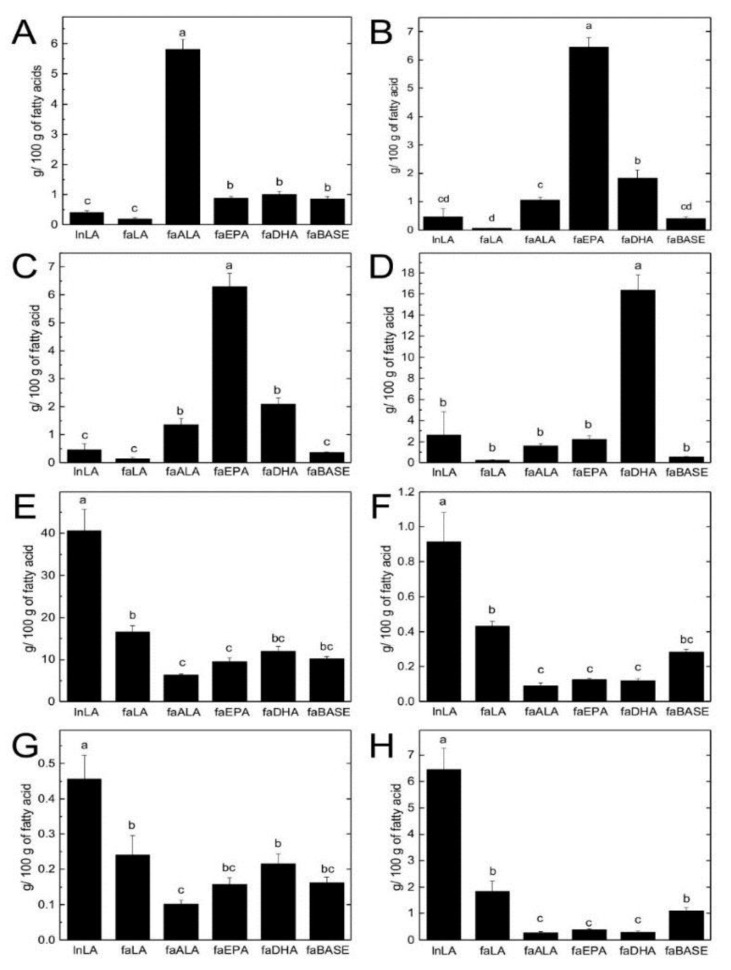
n3 and n6 fatty acids in hepatic triglycerides. ALA (C18:3n3) (**A**), EPA (C20:5n3) (**B**), DPA (C22:5n3) (**C**), DHA (C22:6n3) (**D**), LA (C18:2n6) (**E**), GLA (C18:3n6) (**F**), DGLA (C20:3n6) (**G**), and AA (C20:4n6) (**H**) as g/100 g of fatty acid presented as means ± SEM, *n* = 6/group. Different letters (a,b,c,d) indicate significant differences (*p* < 0.05) among the means. Abbreviations: AA, arachidonic acid; ALA, α-linolenic acid; DGLA, dihommo-gamma-linolenic acid; DHA, docosahexaenoic acid; DPA, docosapentaenoic acid; EPA, eicosahexaenoic acid; faALA, *fa/fa* Zucker rats fed the α-linolenic acid diet; faBASE, baseline *fa/fa* Zucker rats; faDHA, *fa/fa* Zucker rats fed the docosahexaenoic acid diet; faEPA, *fa/fa* Zucker rats fed the eicosahexaenoic acid diet; faLA, *fa/fa* Zucker rats fed the linoleic acid diet; GLA; γ-linolenic acid; lnLA, lean Zucker rats fed the linoleic acid diet.

**Figure 5 nutrients-11-00917-f005:**
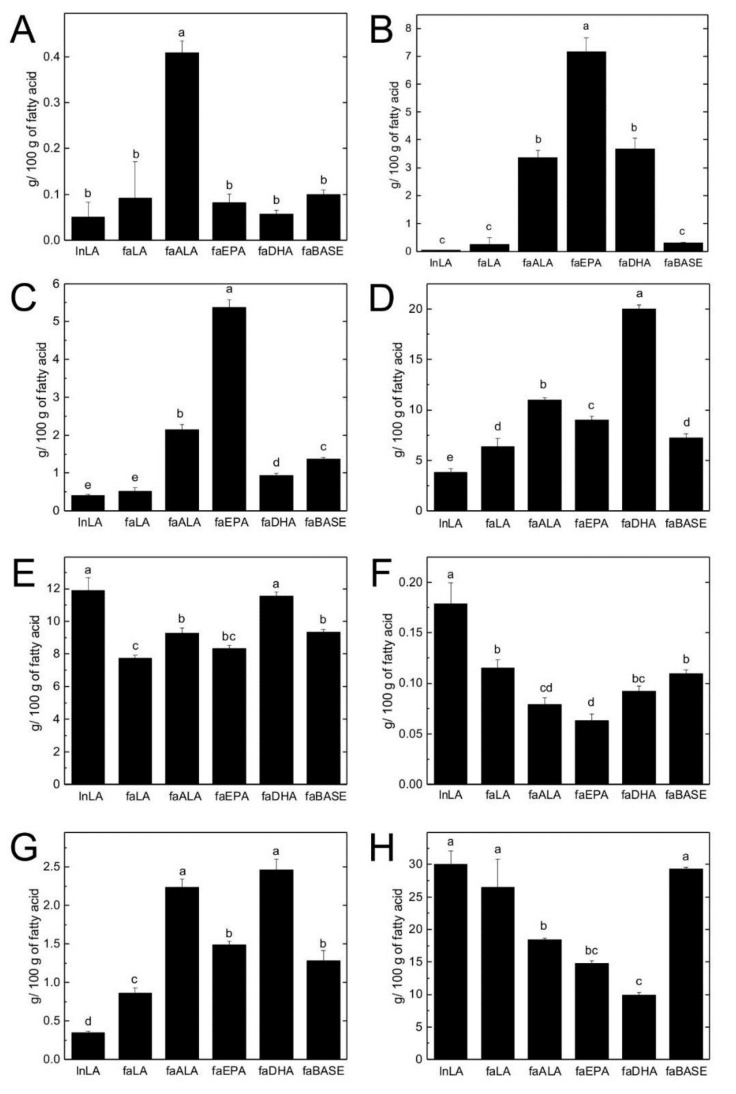
n3 and n6 fatty acids in hepatic phospholipids. ALA (C18:3n3) (**A**), EPA (C20:5n3) (**B**), DPA (C22:5n3) (**C**), DHA (C22:6n3) (**D**), LA (C18:2n6) (**E**), GLA (C18:3n6) (**F**), DGLA (C20:3n6) (**G**), and AA (C20:4n6) (**H**) as g/100 g of fatty acid presented as means ± SEM, *n* = 6/group. Different letters (a,b,c,d,e) indicate significant differences (*p* < 0.05) among the means. Abbreviations: AA, arachidonic acid; ALA, α-linolenic acid; DGLA, dihommo-gamma-linolenic acid; DHA, docosahexaenoic acid; DPA, docosapentaenoic acid; EPA, eicosahexaenoic acid; faALA, *fa/fa* Zucker rats fed the α-linolenic acid diet; faBASE, baseline *fa/fa* Zucker rats; faDHA, *fa/fa* Zucker rats fed the docosahexaenoic acid diet; faEPA, *fa/fa* Zucker rats fed the eicosahexaenoic acid diet; faLA, *fa/fa* Zucker rats fed the linoleic acid diet; GLA; γ-linolenic acid; lnLA, lean Zucker rats fed the linoleic acid diet.

**Table 1 nutrients-11-00917-t001:** Diet formulations.

	LA Diet	ALA Diet	EPA Diet	DHA Diet
**Diet Ingredients (g/kg) ^1^**				
Cornstarch	348	348	348	348
Maltodextrin	132	132	132	132
Sucrose	100	100	100	100
Egg white	213	213	213	213
Cellulose	50	50	50	50
AIN-93G-MX mineral mix	35	35	35	35
AIN-93-VX vitamin mix	10	10	10	10
Choline	3	3	3	3
Biotin mix ^2^	10	10	10	10
Soybean oil	0	0	67	67
High linoleic safflower oil ^3^	100	0	0	0
Flaxseed oil ^4^	0	87	0	0
Canola oil ^5^	0	10	0	0
Coconut oil ^6^	0	3	0	0
EPA oil ^7^	0	0	33	0
DHA oil ^7^	0	0	0	33
**Fatty Acid Composition (g/100 g lipid) ^8^**				
SFA	10	11	10	10
MUFA	17	19	15	15
PUFA	72.3	70	75	75
LA (C18:2n6)	72	18	36	36
ALA (C18:3n3)	0.3	52	6	6
EPA (C22:5n3)	0	0	32	0
DHA (C22:6n3)	0	0	0	33
Other PUFAS	0	0	1	0
n6-PUFA:n3-PUFA	240:1	1:3	1:1.1	1:1.1

^1^ Ingredients from Dyets, Inc (Bethleham, PA) unless otherwise indicated; diets were isocaloric and provided 3.9 kcal/gram. ^2^ 200 mg biotin/kg cornstarch was added because egg white contains avidin which binds to biotin. ^3^ Alnor Oil Company (Valley Stream, NY). ^4^ Omega Nutrition (Vancouver, BC). ^5^ Smuckers Food Services (Markham, ON). ^6^ Nutiva (Richmond, CA). ^7^ Larodan Fine Chemicals (Malmö, SE); >95% purity. ^8^ Determined by gas chromatography as described in the Methods. Abbreviations: AIN, American Institute of Nutrition; ALA, α-linolenic acid; DHA, docosahexaenoic acid; EPA, eicosahexaenoic acid; LA, linoleic acid; MUFA, monounsaturated fatty acids; PUFA, polyunsaturated fatty acids; SFA, saturated fatty acids.

**Table 2 nutrients-11-00917-t002:** Body weight, feed intake, liver weight, and serum biochemistry ^1^.

	lnLA	faLA	faALA	faEPA	faDHA	faBASE
**Body Weight and Feed Intake**						
Initial body weight (g)	218 ± 6 ^c^	323 ± 15 ^a^	316 ± 11 ^a^	323 ± 12 ^a^	297 ± 13 ^a^	247 ± 4 ^b^
Final body weight (g)	423 ± 7 ^c^	663 ± 14 ^a^	635 ± 13 ^a^	593 ± 15 ^b^	568 ± 17 ^b^	---
Total weight gain (g)	211 ± 7 ^d^	357 ± 19 ^a^	336 ± 13 ^ab^	293 ± 14 ^bc^	282 ± 13 ^c^	----
Total feed intake (g)	1219 ± 48 ^c^	1620 ± 38 ^a^	1551 ± 27 ^a^	1392 ± 31 ^b^	1370 ± 76 ^b^	----
Feed efficiency ratio (g total feed intake/g weight gain)	5.80 ± 0.25 ^a^	4.59 ± 0.16 ^b^	4.65 ± 0.13 ^b^	4.80 ± 0.14 ^b^	4.92 ± 0.2 ^b^	----
**Liver Weight**						
Liver weight (g)	13.3 ± 0.7 ^c^	29.0 ± 1.4 ^b^	32.7 ± 1.2 ^a^	28.5 ± 1.2 ^b^	26.0 ± 0.8 ^b^	11.0 ± 0.2 ^d^
Liver weight (g/100 g bwt)	2.9 ± 0.1 ^b^	4.7 ± 0.4 ^a^	5.2 ± 0.2 ^a^	4.8 ± 0.2 ^a^	4.6 ± 0.2 ^a^	4.5 ± 0.1 ^a^
**Serum Biochemistry**						
ALT (U/L)	47 ± 6 ^c^	123 ± 11 ^b^	241 ± 47 ^a^	99 ± 8 ^bc^	94 ± 11 ^bc^	77 ± 3 ^bc^
AST (U/L)	218 ± 17 ^b^	289 ± 15 ^b^	430 ± 58 ^a^	262 ± 20 ^b^	259 ± 32 ^b^	207 ± 18 ^b^
Haptoglobin (mg/mL)	1.51 ± 0.2 ^bc^	2.13 ± 0.2 ^b^	2.12 ± 0.3 ^b^	3.08 ± 0.3 ^a^	3.01 ± 0.2 ^a^	1.12 ± 0.1 ^c^
Total cholesterol (mmol/L)	2.59 ± 0.10 ^c^	7.02 ± 0.66 ^a^	7.29 ± 1.28 ^a^	3.45 ± 0.23 ^bc^	2.79 ± 0.28 ^bc^	4.33 ± 0.23 ^bc^
LDL-C (mmol/L)	0.38 ± 0.03 ^c^	1.56 ± 0.31 ^a^	0.81 ± 0.28 ^b^	0.25 ± 0.04 ^c^	0.17 ± 0.06 ^c^	0.22 ± 0.05 ^c^
HDL-C (mmol/L)	2.39 ± 0.06 ^d^	5.05 ± 0.45 ^a^	3.91 ± 0.48 ^b^	2.36 ± 0.13 ^cd^	1.72 ± 0.23 ^e^	2.94 ± 0.22 ^c^
TG (mmol/L)	0.87 ± 0.09 ^d^	4.66 ± 0.51 ^abc^	5.66 ± 0.53 ^a^	4.02 ± 0.51 ^c^	4.76 ± 0.80 ^bc^	4.81 ± 0.40 ^ab^

^1^ Values are expressed as means ± SEM, *n* = 8–10/group. Different superscript letters (a,b,c,d) within a row indicate significant differences (*p* < 0.05) among the means. Abbreviations: ALT, alanine aminotransferase; AST, aspartate aminotransferase; bwt, body weight; faALA, *fa/fa* Zucker rats fed the α-linolenic acid diet; faBASE, baseline *fa/fa* Zucker rats; faDHA, *fa/fa* Zucker rats fed the docosahexaenoic acid diet; faEPA, *fa/fa* Zucker rats fed the eicosahexaenoic acid diet; faLA, *fa/fa* Zucker rats fed the linoleic acid diet; HDL-C, high density lipoprotein-cholesterol; LDL-C, low density lipoprotein-cholesterol; lnLA, lean Zucker rats fed the linoleic acid diet; TG, triglycerides.

**Table 3 nutrients-11-00917-t003:** Hepatic TG and PL fatty acid composition ^1^.

	lnLA	faLA	faALA	faEPA	faDHA	faBASE
**TG Fraction**						
Total SFA	27.3 ± 2.5 ^d^	40.7 ± 1.4 ^ab^	42.1 ± 0.9 ^ab^	39.1 ± 0.9 ^bc^	31.9 ± 5.2 ^cd^	46.4 ± 0.7 ^a^
Total MUFA	18.4 ± 1.0 ^d^	37.3 ± 0.8 ^bc^	41.2 ± 0.6 ^a^	34.5 ± 1.1 ^c^	33.0 ± 2.4 ^c^	39.4 ± 1.1 ^ab^
Total PUFA	54.3 ± 3.3 ^a^	20.6 ± 2.2 ^d^	16.8 ± 0.9 ^de^	26.4 ± 2.1 ^c^	34.1 ± 3.1 ^b^	14.1 ± 0.9 ^e^
Total n3-PUFA	3.64 ± 2.5 ^c^	0.37 ± 0.1 ^c^	8.77 ± 0.6 ^b^	8.65 ± 0.8 ^b^	19.5 ± 1.8 ^a^	1.75 ± 0.2 ^c^
Total n6-PUFA	50.6 ± 5.7 ^a^	20.3 ± 2.2 ^b^	7.99 ± 0.3 ^d^	17.7 ± 1.3 ^bc^	14.6 ± 1.4 ^bcd^	12.4 ± 0.8 ^cd^
n6/n3 Ratio	45.2 ± 11.4 ^b^	64.2 ± 7.4 ^a^	0.92 ± 0.03 ^c^	2.09 ± 0.1 ^c^	0.75 ± 0.05 ^c^	7.20 ± 0.3 ^c^
**Phospholipid Fraction**						
Total SFA	43.9 ± 0.1	46.0 ± 0.5	45.3 ± 0.7	45.5 ± 0.6	44.4 ± 0.4	43.5 ± 0.1
Total MUFA	5.89 ± 0.1	9.05 ± 2.5	6.62 ± 0.2	7.25 ± 0.2	6.07 ± 0.2	7.22 ± 1.9
Total PUFA	50.1 ± 0.1 ^a^	45.0 ± 2.6 ^b^	48.0 ± 0.6 ^ab^	47.2 ± 0.5 ^ab^	50.0 ± 0.3 ^a^	49.3 ± 1.6 ^a^
Total n3-PUFA	8.7 ± 0.4 ^d^	12.8 ± 5 ^bc^	17.4 ± 3.6 ^ab^	15.2 ± 0.3 ^abc^	21.5 ± 0.4 ^a^	4.85 ± 0.4 ^cd^
Total n6-PUFA	41.4 ± 0.4 ^a^	32.2 ± 5.6 ^b^	30.6 ± 3.5 ^b^	32.0 ± 0.2 ^b^	28.0 ± 0.4 ^b^	44.5 ± 1.4 ^a^
n6/n3 Ratio	4.81 ± 0.2 ^b^	4.36 ± 1.1 ^b^	2.11 ± 0.4 ^c^	2.10 ± 0.03 ^c^	1.30 ± 0.04 ^c^	9.39 ± 0.6 ^a^

^1^ g/100 g fatty acids, except for the ratios. Values are expressed as means ± SEM, *n* =6/group. Different superscript letters (a,b,c,d,e) with a row indicate significant differences (*p* <0.05) among the means and an absence of letters within a row indicates no significant differences. Abbreviations: faALA, *fa/fa* Zucker rats fed the α-linolenic acid diet; faBASE, baseline *fa/fa* Zucker rats; faDHA, *fa/fa* Zucker rats fed the docosahexaenoic acid diet; faEPA, *fa/fa* Zucker rats fed the eicosahexaenoic acid diet; faLA, *fa/fa* Zucker rats fed the linoleic acid diet; lnLA, lean Zucker rats fed the linoleic acid diet; MUFA, monounsaturated fatty acids; PL, phospholipid fraction; PUFA, polyunsaturated fatty acids; SFA, saturated fatty acids; TG, triglyceride fraction.

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
