# Peer review of "Dietary Docosahexaenoic Acid (DHA) and Eicosapentaenoic Acid (EPA) Operate by Different Mechanisms to Modulate Hepatic Steatosis and Hyperinsulemia in fa/fa Zucker Rats"

_nutrients, 2019, doi:10.3390/nu11040917_

Reviewer 1 Report

The authors investigated the effects of n3-PUFA on hepatic steatosis, markers of hepatic inflammation and fibrosis, and insulinemia in fa/fa Zucker rats. Among the n3-PUFA, DHA most effectively prevented the progression of hepatic steatosis via the reduction of FAS expression. This manuscript provides new insights into the physiological functions of n3-PUFA. However, to improve this manuscript, the authors should address the following points.

 Major comments:

1. Although hepatic lipid droplet assessment showed no differences in the average size of lipid droplets among the fa/fa experimental groups (Figure 1B), the authors described the size distribution of the lipid droplets (in page 5, line 182-194 and Figure 1C). The authors should show the significant difference of size distribution and, on the base of the difference, discuss the size distribution among the fa/fa experimental groups.

 2. The hepatic lipid concentration of the faEPA and faDHA groups were decrease (Figure 1A) and the faEPA and faDHA groups had small lipid droplets (Figure 1C). Moreover, FAS expression levels in these groups were decreased (Figure 1D). Therefore, the authors conclude that the decrease of FAS levels is associated with the reduction of hepatic steatosis. However, FAS was highly expressed in faBASE groups, although the lipid drop size is very small. The authors should explain this point.

 Minor comments:

1. In page 9, line 269-270, the authors mentioned “The faEPA had …. and faBASE groups”. However, TG concentration in lnLA is lowest. The authors should correct the sentence.

 2. The authors should show the information (year and page etc.) of reference 11.

Author Response

 Major comments:

Although hepatic lipid droplet assessment showed no differences in the average size of lipid droplets among the fa/fa experimental groups (Figure 1B), the authors described the size distribution of the lipid droplets (in page 5, line 182-194 and Figure 1C). The authors should show the significant difference of size distribution and, on the base of the difference, discuss the size distribution among the fa/fa experimental groups.

Response:  We have added some more explanation to describe the changes among the groups for the different size categories. Both the Results text and the Figure 1 legend have the statement:  “Chi-squared testing confirmed that there were differences among the dietary groups for all the size ranges.” Based on consultations with our statistician, Chi-squared testing informs that there are differences among the groups, but there is no additional statistical test to distinguish differences between specific groups. The proposed differences can be provided descriptively as we have done in the text.

The hepatic lipid concentration of the faEPA and faDHA groups were decrease (Figure 1A) and the faEPA and faDHA groups had small lipid droplets (Figure 1C). Moreover, FAS expression levels in these groups were decreased (Figure 1D). Therefore, the authors conclude that the decrease of FAS levels is associated with the reduction of hepatic steatosis. However, FAS was highly expressed in faBASE groups, although the lipid drop size is very small. The authors should explain this point.

Response:  We did not quantify lipid droplet area or distribution for the faBASE group (Figure 1B, 1C) thus we cannot address the point raised by the reviewer.  The faLA and faALA groups had elevated levels of FAS and more lipid droplets in the larger size categories compared to the faEPA and faDHA groups.

 Minor comments:

In page 9, line 269-270, the authors mentioned “The faEPA had …. and faBASE groups”. However, TG concentration in lnLA is lowest. The authors should correct the sentence.

Response:  The sentence has been corrected:  “The faEPA had lower serum TG concentrations than the faBASE but not the lnLA group.”

The authors should show the information (year and page etc.) of reference 11.

Response:  This information has been provided for reference 11.

Reviewer 2 Report

This is an interesting manuscript investigating the effect of individual PUFA (LA, ALA, EPA and DHA) on NAFLD progression and comorbidities in hyperphagic rats. The authors’ approach is interesting. Instead of merely using a supplementation of PUFA to the diet, a significant proportion of all fatty acids in the diet are PUFA. A recent meta-analysis study by Gao H et al., 2017, has shown that simple PUFA (fish oil) supplementation is ineffective in treating TD2 in human. In this study, they also found that while short term supplementation improves insulin sensitivity, it is not observed at long term.

The present study is interesting because it evaluates the effect that a more drastic change in diet could have on obese individuals as well as the effect of individual PUFA on obese rats. While the findings are interesting, I have identified issues that must be addressed.

Major issues:

1-        The faBASE group was sacrificed at the age of 6 weeks, one week after acclimation which is the precise time at which the other group entered the 8 weeks diet protocol. The initial weight of the lnLA, faLA, faALA, faEPA and faDHA was also taken before the 8 weeks diet protocol. Thus, the weight of faBASE rats at that time point should be placed in the initial weight (g) line in table 2 instead of the final weight line.

 This modification highlights an important problem. When comparing the initial weight of the animals, the control group faBASE have a lower body weight compared to every other fa groups even before the diet protocol has started! This was verified using GraphPad Prism 7 software using the data provided in Table 2 with one-way ANOVA and Tukey’s multiple comparison test (see below). This would indicate that the groups formation was not performed appropriately. All obese Zucker rat groups should have started with the same average body weight. This error introduces a bias in this study that must be addressed.

Tukey's   multiple comparisons test

Mean   Diff.

95.00%   CI of diff.

Significant?

Summary

Adjusted   P Value

lnLA vs. faLA

-105

-151 to -59.04

Yes

****

<0.0001< p="">

A-B

lnLA vs. faALA

-98

-144 to -52.04

Yes

****

<0.0001< p="">

A-C

lnLA vs. faEPA

-105

-151 to -59.04

Yes

****

<0.0001< p="">

A-D

lnLA vs. faDHA

-79

-125 to -33.04

Yes

****

<0.0001< p="">

A-E

lnLA vs. faBASE

-29

-74.96 to 16.96

No

ns

0.4260

A-F

faLA vs. faALA

7

-38.96 to 52.96

No

ns

0.9974

B-C

faLA vs. faEPA

0

-45.96 to 45.96

No

ns

>0.9999

B-D

faLA vs. faDHA

26

-19.96 to 71.96

No

ns

0.5465

B-E

faLA   vs. faBASE

76

30.04   to 122

Yes

***

0.0002

B-F

faALA vs. faEPA

-7

-52.96 to 38.96

No

ns

0.9974

C-D

faALA vs. faDHA

19

-26.96 to 64.96

No

ns

0.8176

C-E

faALA   vs. faBASE

69

23.04   to 115

Yes

***

0.0008

C-F

faEPA vs. faDHA

26

-19.96 to 71.96

No

ns

0.5465

D-E

faEPA   vs. faBASE

76

30.04   to 122

Yes

***

0.0002

D-F

faDHA   vs. faBASE

50

4.043   to 95.96

Yes

*

0.0260

E-F

1-      In figure 2, most of the fa groups are at the same level as the lnLA groups. This suggest that all PUFA tested except ALA can reverse early fibrogenesis? While aSMA is a good biomarker for HSCs activation, it is only one marker. This part of the study should be expanded to at least a second biomarker such as COL1A1 or by using a histological approach such as Sirius Red or Masson’s trichrome coloration to better quantify fibrosis in the liver of these rats.

However, it should be noted that the fibrosis results on Zucker rats (fa/fa) can hardly be translated to humans or other rodent models. This is because fa/fa rats are resistant to fibrosis due to a lack of leptin signaling (Ikejima K et al. 2002), (Vivoli E et al. 2016). This point should be acknowledged by the author.

 2-      In figure 3, the author shows the insulin sensitivity parameters and the effect on FOXO1. However, there seems to be no correlation between HOMA-IR index (or insulin levels) and FOXO1 levels. A fact that seems to be ignored by the author. According to panel C, the faEPA rats have a reduced sensibility to insulin, however they express FOXO1 at the same level as the healthy lnLA rats. The author should at least measure the activation of AKT by phosphorylation to try to make sense of this data and to have at least some measure of insulin sensitivity.

 3-      The author suggests that the observed effects of faDHA treatments might be linked to a reduction of liver inflammation but they only measured serum haptoglobin (a non-specific marker of systemic inflammation) as a marker. Furthermore, the author implies that arachidonic acid, the precursor of all eicosanoids (molecules deeply involved in inflammation stimulation and resorption) is also implicated. While the exact molecular pathways implicated could be explored later in another study, the author should at least determine the inflammatory state of these rat’s liver for the present study. The liver mRNA expression of a few cytokines and inflammation markers should be measured. Tnfa, Il6, Il1b. Dd68 and/or Adgre1 mRNA levels or the nuclear recruitment of NF-kB could be assessed. Inflammation is a strong inhibitor of insulin signaling. Determining the inflammatory state of the liver could help to tie all the authors data together.    

 Minor issues:

1-      On line 43: The phrasing is strange. Did the author meant: Patients with hepatic steatosis have a lower relative concentration of n3-PUFA in blood and liver tissue biopsies?

2-      The lean Zucker rats are heterozygotes for the leptin receptor gene defect (FA/fa, if fa is the mutant allele and FA the WT (+) allele) or are they WT homozygotes (FA/FA)? The exact genotype used in this study should be clearly stated in the material and methods when first referring to the lean Zucker Rats.

3-      Upon inspection, the caloric density (kCal/kg of food) of each diet seems to be roughly equal between each diet. The author should confirm this and include this data in table 1.

4-      In table 2, the author should add the liver weight without normalisation. While the liver weight index can be useful it should always be accompanied by the net liver weight. In figure 1, the author shows that the faDHA group have a lower concentration of lipids in their livers compared to the other fa groups after the diet protocol. However, the liver weight index is not impacted in table 2. This could imply that while the lipid content is lower in this group liver weight is equal to other fa groups, meaning that the liver weight is attributable to another parameter or that liver index is hiding the effect that could be observed on net liver weight.

Author Response

Major issues:

1-      In figure 2, most of the fa groups are at the same level as the lnLA groups. This suggest that all PUFA tested except ALA can reverse early fibrogenesis? While aSMA is a good biomarker for HSCs activation, it is only one marker. This part of the study should be expanded to at least a second biomarker such as COL1A1 or by using a histological approach such as Sirius Red or Masson’s trichrome coloration to better quantify fibrosis in the liver of these rats.

However, it should be noted that the fibrosis results on Zucker rats (fa/fa) can hardly be translated to humans or other rodent models. This is because fa/fa rats are resistant to fibrosis due to a lack of leptin signaling (Ikejima K et al. 2002), (Vivoli E et al. 2016). This point should be acknowledged by the author.

Response:  We thank the reviewer for the valuable suggestions, however, given the 1 week response time expected by Nutrients we had to focus our efforts on analyses for which we had antibodies and supplies on hand. We did not have the COL1A1 antibody on hand or the time to do more sectioning and staining of the liver.  However, we have been able to add some new data with respect to points 2 and 3 (see below). We appreciate the information and references indicating that ‘fa/fa rats are resistant to fibrosis due to a lack of leptin signaling ‘(Ikejima K et al. 2002)’ and we have pointed this out in the Discussion: “However, it needs to be noted that others have reported that fa/fa Zucker rats are resistant to fibrosis induced chemically or with a Western diet (based on mRNA levels of α-SMA and histological assessment) as leptin receptor-mediated signaling appears to play a role in development of hepatic fibrosis and remodeling of the extracellular matrix (Ikejima 2002; Saito 2018). Further histological assessment and additional fibrotic markers are required for a more in-depth understanding of the reduction in hepatic α-SMA with DHA intervention in the current study.”

2-      The author suggests that the observed effects of faDHA treatments might be linked to a reduction of liver inflammation but they only measured serum haptoglobin (a non-specific marker of systemic inflammation) as a marker. Furthermore, the author implies that arachidonic acid, the precursor of all eicosanoids (molecules deeply involved in inflammation stimulation and resorption) is also implicated. While the exact molecular pathways implicated could be explored later in another study, the author should at least determine the inflammatory state of these rat’s liver for the present study. The liver mRNA expression of a few cytokines and inflammation markers should be measured. Tnfa, Il6, Il1b. Dd68 and/or Adgre1 mRNA levels or the nuclear recruitment of NF-kB could be assessed. Inflammation is a strong inhibitor of insulin signaling. Determining the inflammatory state of the liver could help to tie all the authors data together.    

 Response:   We thank the reviewer for these suggestions.  We do not have primers on hand to analyze the mRNA levels of Tnfa, Il6, Il1b, Dd68 and/or Adgre1 within the timeframe required by Nutrients for revisions.  However, we were able to do Western blotting for F4/80 and p-NF-kB p65 (Figure 2A, 2B). This has yielded some interesting new data which have been incorporated into the Results and Discussion. We have been able to expand our conclusions that n3-PUFA and n6-PUFA differentially regulate the various inflammatory pathways.

We appreciate the efforts of the reviewer and these additions have enhanced the quality of the manuscript.

Minor issues:

On line 43: The phrasing is strange. Did the author meant: Patients with hepatic steatosis have a lower relative concentration of n3-PUFA in blood and liver tissue biopsies?

 Response: We have re-worded the sentence as suggested.

 The lean Zucker rats are heterozygotes for the leptin receptor gene defect (FA/fa, if fa is the mutant allele and FA the WT (+) allele) or are they WT homozygotes (FA/FA)? The exact genotype used in this study should be clearly stated in the material and methods when first referring to the lean Zucker Rats.

 Response: We have clarified in the Methods that the lean Zucker rats are +/? as they were not genotyped.

 Upon inspection, the caloric density (kCal/kg of food) of each diet seems to be roughly equal between each diet. The author should confirm this and include this data in table 1.

 Response: We have included in footnote 1 of Table 1 that the diets were isocaloric and provided 3.9 kcal/gram diet

 In table 2, the author should add the liver weight without normalisation. While the liver weight index can be useful it should always be accompanied by the net liver weight. In figure 1, the author shows that the faDHA group have a lower concentration of lipids in their livers compared to the other fa groups after the diet protocol. However, the liver weight index is not impacted in table 2. This could imply that while the lipid content is lower in this group liver weight is equal to other fa groups, meaning that the liver weight is attributable to another parameter or that liver index is hiding the effect that could be observed on net liver weight.

Response:  The absolute liver weight has been added to Table 2, and a sentence describing this parameter has been added to the Results:  “Absolute liver weight was greater in the faALA group compared to the faLA, faEPA and faDHA groups, and 2-fold higher in the 14 week old fa/fa rats compared to the lnLA group (Table 2).” In the Discussion we have added:  “The parallel changes in absolute liver weight and liver lipid content support the evidence that DHA and EPA reduced hepatic lipid deposition.”

Round  2

Reviewer 1 Report

My concerns were addressed.

Author Response

Reviewer 1:  My concerns were addressed.

Response:  Thank you.  We appreciated your constructive comments.
Reviewer 2 Report

1-    I have reviewed the new version of the proposed manuscript. I appreciate the effort that the authors have deployed to enhance the quality of their article. I believe the added experiment sufficiently clarify the haziest elements of the previous version. I would have liked to see the mRNA expression of inflammatory marker to be certain of the inflammatory state of these rats livers but the short amount of time given to the author would not have been enough to provide those. While P65 phosphorylation is an indicator of upstream stress leading to inflammation, other pathways dependent on PUFA sensing (PPARγ and others) acts downstream of p65 to limit inflammation. Fortunately, F4/80 was also included. F4/80 is an indicator of macrophage invasion and proliferation in the liver. This is a better indicator of general liver inflammation. We know that inflammation correlates negatively with insulin sensitivity. The inclusion of F4/80 is fortunate because it correlates well with insulin levels, HOMA-IR indexes.

        2-     ‘’Furthermore, consumption of n6-PUFA did not have a pro-inflammatory effect in the context of elevated hepatic lipid as markers of inflammation (haptoglobin, F4/80, p-NF-κB p65) and hepatic function (ALT, AST) were not increased in the faLA group relative to baseline or lean Zucker rats.’’

Can this really be said with the authors data? There is no control chow diet. In this case, the lean control is also nourished with omega-6 fatty acids… If LA has a pro-inflammatory effect, it would not be possible to evaluate with this experimental design. Rather it could be said that omega-6 fatty acid diet did not worsen the initial inflammatory state of the faBASE group. The faLA have a higher Haptaglobin levels than faBASE and are marginaly higher than lnLA. I would give more credence to F4/80 than to NF-kB as a reliable marker of inflammation. Considering this, all the fa groups would have higher inflammation levels compared to the lnLA healthy rats with slightly less inflammation in faLA, faEPA and faDHA compared to faBASE and faALA. 

3-    I also appreciate the inclusion of p-AKT and pAMPK as markers. AKT phosphorylation does indicate that insulin signaling is affected in fa/fa rats. Surprisingly, dietary treatment with PUFAs seems to have no effect on these levels despite having a net effect on insulin sensitisation. Maybe the effect is downstream of AKT? I have only a minor complaint for this part of the study. The inclusion of the pAKT and pAMPK in figure 1 doesn’t make sense. It should instead be included in figure with the other insulin data.

       I am satisfied with the other changes.

Author Response

1-    I have reviewed the new version of the proposed manuscript. I appreciate the effort that the authors have deployed to enhance the quality of their article. I believe the added experiment sufficiently clarify the haziest elements of the previous version. I would have liked to see the mRNA expression of inflammatory marker to be certain of the inflammatory state of these rats livers but the short amount of time given to the author would not have been enough to provide those. While P65 phosphorylation is an indicator of upstream stress leading to inflammation, other pathways dependent on PUFA sensing (PPARγ and others) acts downstream of p65 to limit inflammation. Fortunately, F4/80 was also included. F4/80 is an indicator of macrophage invasion and proliferation in the liver. This is a better indicator of general liver inflammation. We know that inflammation correlates negatively with insulin sensitivity. The inclusion of F4/80 is fortunate because it correlates well with insulin levels, HOMA-IR indexes.

Response:  Thank you for your comments and pointing out the value of including F4/80.
 2-     ‘’Furthermore, consumption of n6-PUFA did not have a pro-inflammatory effect in the context of elevated hepatic lipid as markers of inflammation (haptoglobin, F4/80, p-NF-κB p65) and hepatic function (ALT, AST) were not increased in the faLA group relative to baseline or lean Zucker rats.’’

Can this really be said with the authors data? There is no control chow diet. In this case, the lean control is also nourished with omega-6 fatty acids… If LA has a pro-inflammatory effect, it would not be possible to evaluate with this experimental design. Rather it could be said that omega-6 fatty acid diet did not worsen the initial inflammatory state of the faBASE group. The faLA have a higher Haptaglobin levels than faBASE and are marginaly higher than lnLA. I would give more credence to F4/80 than to NF-kB as a reliable marker of inflammation. Considering this, all the fa groups would have higher inflammation levels compared to the lnLA healthy rats with slightly less inflammation in faLA, faEPA and faDHA compared to faBASE and faALA.

 Response:  Thank you for pointing out the inaccuracies in our statements regarding n6-PUFA and inflammation.  We have revised statements in the manuscript as follows:

Line 378-381: “Furthermore, consumption of n6-PUFA did not have a pro-inflammatory effect in fa/fa Zucker rats in the context of elevated hepatic lipid as markers of inflammation (F4/80, p-NF-κB p65) and hepatic function (ALT, AST) were not increased in the faLA group relative to baseline Zucker rats.”

Lines 508-511: “However, in the present study, n6-PUFA did not worsen the initial inflammatory state of genetically obese rats as markers of inflammation (F4/80, p-NF-κB p65) and hepatic function (ALT, AST) were not increased in the faLA group relative to baseline fa/fa Zucker rats.”

3-    I also appreciate the inclusion of p-AKT and pAMPK as markers. AKT phosphorylation does indicate that insulin signaling is affected in fa/fa rats. Surprisingly, dietary treatment with PUFAs seems to have no effect on these levels despite having a net effect on insulin sensitisation. Maybe the effect is downstream of AKT? I have only a minor complaint for this part of the study. The inclusion of the pAKT and pAMPK in figure 1 doesn’t make sense. It should instead be included in figure with the other insulin data.

Response:  We have moved pAKT and pAMPK to Figure 3 (Panels E and F).  The figure legends and text (Lines 272-275) have been adjusted accordingly.
I am satisfied with the other changes.

Response:  Thank you for all your constructive comments. The various additions and corrections have enhanced the quality of our manuscript.